

# Inversion of vertical mass concentration of non-spherical aerosols using multi-wavelength lidar

Hu Zhao[1*, 2], Ze Qiao[1, 2], Jiyuan Cheng[1, 2] Jiandong Mao[1, 2] Chunyan Zhou[1, 2] Xin Gong[1, 2] Zhiming Rao[1, 2]

[1]School of Electrical and Information Engineering, North MinZu University, Yinchuan, 750021,China
[2]Key Laboratory of Atmospheric Environment Remote Sensing, Ningxia, 750021, China

*Correspondence to*: Hu Zhao (zhaoh_1@yeah.net)

**Abstract.** A method of inversion the vertical mass concentration of non-spherical aerosols is proposed in this paper and the effect of particle shape, complex refraction index and wavelength on the inversion is investigated. The experiment of multi-

wavelength polarized lidar indicates that the aerosols with depolarization ratios of 0.22-0.37 have non-spherical characteristics. Given the non-spherical character of dust aerosols, the optical properties with different shapes, wavelengths, and complex refraction index are calculated using discrete dipole approximation. Based on previous studies, dust aerosol particles were assumed to be ellipsoidal and rectangular in this study. In several non-spherical shapes we studied, it found that when the shape parameter of non-spherical particles is $D$=1 rectangular, the mass concentration difference between non-

spherical and spherical aerosol is the largest, and the maximum difference can reach 19.08%. It was found that in the detection of aerosol mass concentration by multi-wavelength lidar, the larger the wavelength, the smaller the aerosol mass concentration. It was also found that the larger the real part of the complex refraction index, the smaller the mass concentration and the smaller the maximum difference between the three wavelengths. The vertical mass concentrations inversed using this method are 2.83mg/m$^3$, 2.51mg/m$^3$, and 4.2mg/m$^3$ at 2:03, 7:28, and 12:36 on March 16, 2021, at the

altitude of 1.5km respectively. The inverted vertical mass concentrations were compared with the near-ground monitored at the observatory. The results show that there is same correspondence between the changes in aerosol mass concentrations near the ground and at high altitudes at the same moment.

## 1 Introduction

The frequent ravages of dusty weather in recent years have had a serious impact on global climate change (Meo et al., 2021).

Dust aerosols are the main components of springtime aerosols in Asia (Ge et al., 2023). Dust causes adverse effects in terms of vegetation, transportation safety, and air quality. The mass concentration of dust aerosols can be used to assess the extent of particulate pollution in the air (Zhang et al., 2024). Therefore, by studying the mass concentration of dust aerosols, the state of the atmosphere and the associated human health problems can be better assessed. This can contribute to the protection of human health and sustainable development.



It is worth noting that dust aerosols are non-spherical particles (Kong et al., 2022). Non-spherical particles may have ellipsoidal, rectangular, or other irregular shapes (Saito et al., 2022). However, in the past, when calculating the optical properties of dust aerosols, the Mie method is usually used to calculate the dust aerosols assuming that the dust aerosols are spherical particles (P érez et al., 2006; Wang et al., 2020; Abuelgasim et al., 2020; Mie et al., 1908). The Mie method assumes that the dust aerosol is a spherical particle, which brings great error to the calculation of light scattering parameters of non-spherical particles. It is obviously not suitable. Currently, the commonly used methods for calculating the optical

properties of non-spherical aerosols include the Finite Difference Time Domain (FDTD), the T-matrix method, and the Discrete Dipole Approximation (DDA), etc. (Ishimoto et al., 2019; Tang et al., 2023; Cheremisin et al., 2022). FDTD is more effective in the simulation of complex structures and inhomogeneous media, but FDTD requires larger computational resources because the simulation of non-spherical particles requires meshing of the particles (Yee et al., 1966). T-matrix is a

method based on the matrix description of the scattering properties (Harada et al., 1996), which is mainly used for rotationally symmetric non-spherical particles, but is not effective enough for the calculation of particles with complex shapes. DDA discretizes the particle into a series of small dipoles. It can simulate a variety of particle shapes and can calculate scattering effects within the particles (Fiveland et al., 1987). In this study, when we want to calculate the optical properties of rectangular particles, it is more appropriate to use the DDA method. DDA has been used in the study of non-

spherical aerosols with good results (Jang et al., 2022; Moradi et al., 2022), so in this study, the DDA method is used to calculate the optical properties of non-spherical particles.

Currently, there are three main categories of methods for calculating mass concentrations: the first is point detection methods. Point measurement methods primarily utilize point measurement equipment for the detection of mass concentrations (Kokkalis et al., 2017; Mamali et al., 2018; Wang et al., 2019). However, these devices can only realize single-point

monitoring and cannot acquire the spatial variation of aerosols mass concentrations. The second category is satellite passive remote sensing measurement methods. This type of method can monitor the spatial and temporal distribution changes of aerosols mass concentration over a large area (Chen et al., 2013; Ma et al., 2014; Liu et al., 2015; Yao et al., 2019; Zang et al., 2018). However, this method is unable to calculate the detailed characteristics of the vertical distribution of mass concentration. The third category is the lidar remote sensing measurement method, and the research on lidar based mass

concentration profile detection and inversion methods has attracted wide attention in recent years (Yukari et al., 2018; Siomos et al., 2017; Miatselskaya et al., 2016; Tao et al., 2016; Li et al., 2021). This method allows for a wide range of spatial distributions of vertically-resolved observation, as well as continuous observation of dusty or hazy weather. However, the effect of the optical properties on the mass concentration of non-spherical aerosols is currently subject to further study.

An improved method for calculating the mass concentration of dust aerosols is proposed in this paper. In this method,

aerosol mass concentrations can be obtained by combining DDA and aerosol particle size distribution (APSD) calculations. Among them, the APSD can be obtained by using principal component analysis or the regularization method (Donovan et al., 1997; M üller et al., 1999). Dust aerosols are detected using Multi-wavelength polarized lidar. The optical coefficients of the aerosols were inverted based on the received signals. Additionally, the APSD was reversed using the regularization method.



The particle mass extinction efficiency is obtained by APSD and DDA. Vertical profiles of particle mass concentration were
obtained by combining the profiles of aerosol extinction coefficient. The applicability of the dust aerosol mass concentration
inversion is estimated based on the linear relationship between extinction coefficient and mass concentration. This paper also
discusses the effect of non-spherical particle optical properties on dust mass concentrations, such as shape, wavelength, and
complex refraction index (CRI). The vertical distribution of dust aerosol mass concentrations is better understood as a result
of this work.

## 2 Experimental equipment and inversion methods

### 2.1 Multi-wavelength polarized lidar

In March 2021, dust storms were detected using a Multi-wavelength polarized lidar system in this study. The experimental
site is located on 106.107 °E, 38.498 °N. The multi-wavelength polarization lidar can continuously detect the changes of
aerosols in the vertical atmosphere for a long time. The laser wavelengths include 355nm, 1064nm, and 532nm (polarization
channel). 532nm polarized signals include 532S vertical and 532P parallel. Different wavelengths are sensitive to different
particle radius. Therefore, the optical parameters of different wavelengths can well reflect the distribution of aerosol particles
of different sizes. The extinction coefficient $\alpha(z)$ and backscattering coefficient $\beta(z)$ were derived using the Klett method
inversion (Fernald et al., 1984; Klett et al., 1981). Table 1 displays the parameters of Multi-wavelength polarized lidar.

**Table 1** Configuration of Multi-wavelength polarized lidar

| Laser type | Nd: YAG |
|---|---|
| Wavelength (nm) | 355, 532, 1064 |
| Typical repetition rate (Hz) | 10 |
| Pulse duration (ns) | 10 |
| Extinction ratio of laser | >100:1 |
| Telescope type | Cassegrain |
| Primary mirror diameter (mm) | 250 |
| Field of view (mrad) | 0.6 |
| Detector | PMT (355nm, 532nm), APD (at 1064nm) |
| Channel separation | Dichroic mirror, polarization beam splitter (PBS) and interferential filters (IF) |


### 2.2 Discrete Dipole Approximation

The Discrete Dipole Approximation (DDA) is a computational method of light scattering using dipole equivalent scatterers
to model arbitrarily shaped particles. It is assumed that an initial target volume is formed by $N$ dipoles spaced at $d$. Each
dipole represents the dipole moment of a particular sub volume of the target volume. The higher the number of dipoles in the
target volume, the more accurate the calculation results will be. The number of dipoles $N$=63461. The dipole spacing is
adjusted by $d=(V/N)^{1/3}$, so that the original target volume can be defined as $V=Nd^3$. The equation for the electric dipole
moment $\mathbf{P}_j$ is shown in Eq. (1) (Yurkin et al., 2007):





$$\mathbf{P}_j = \alpha_j \mathbf{E}_j = \alpha_j \left( \mathbf{E}_{j,inc} - \sum_{i=1, j\neq i}^{N} \mathbf{A}_{ij} \cdot \mathbf{p}_i \right) \tag{1}$$

Where $\mathbf{P}_j$ is the instantaneous electric dipole moment of dipole $j$, and where $\mathbf{E}_j$ represents the electric field located at $\mathbf{r}_j$,

consisting of the electric field of the incident field at $j$ and the fields generated by the dipoles at $j$ for all other $i$ locations. The value of the electric field is related to the incident wave and the scattering value of the $N$-1 dipoles. $\alpha_j$ represents the polarizability of the electric dipole with an electric dipole located at $\mathbf{r}_j$.

The spatial orientation of the particles is randomly distributed in the scatter. Thus, the light scattering parameter in this research is represented by the average of the optical values of the particles in each direction. $Q$ is the optical characteristic

parameter, the average value of which is denoted as $\langle Q \rangle$, and the expression is given in Eq. (2) (Yurkin et al., 2007):

$$\langle Q \rangle = \frac{1}{8\pi^2} \int_0^{2\pi} \int_{-1}^{1} \int_0^{2\pi} Q(\beta, \theta, \phi) d\beta d\cos\theta d\phi \tag{2}$$

$$Q_{ext} = \frac{C_{ext}}{\pi a_{\ell f}^2} = \frac{4k}{\left( a_{eff}^2 \left| \vec{E}_0 \right|^2 \right)} \sum_1^N \text{Im}\left( \vec{E}_{inc,j}^* \cdot \vec{p}_i \right) \tag{3}$$

$Q_{ext}$ is the extinction efficiency factor, $P_i$ is the electric dipole moment of the dipole at $i$ ($i \neq j$). $Q(\beta,\theta,\phi)$ is the value of the parameter of the radiative properties of the non-spherical particles at a particular orientation in the system coordinates. $\alpha$ is

the dipole polarizability, $\lambda$ is the incident wavelength, and $N$ is the number of dipoles. $r$ is the radius of the equivalent spherical shape of the non-spherical particle. $x$ is the size parameter. $x=k*r$, $k=2\pi/\lambda$. Considering the radius of dust aerosol particles is mainly concentrated in 0.01-3μm, $r$ is calculated in the range of 0.01-3μm (Müller et al., 1999).

The shape parameter $D$ indicates the degree of non-sphericity, which makes it easier to describe non-spherical aerosol particles. The degree of ellipsoid deviation from the sphere can be accurately described as $D=a/b$, where the short axis is

represented by $a$, and the long axis is represented by $b$.

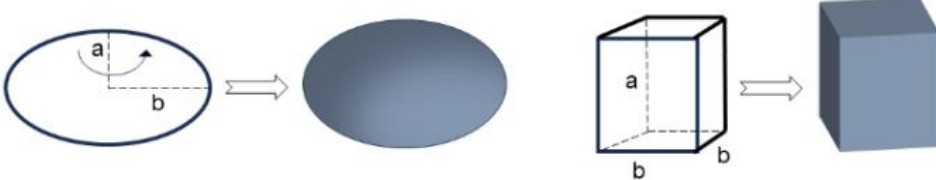

**Figure 1: Schematic diagram of a non-spherical particle**

Fig. 1 illustrates that when $D = 1$, it is spherical. The particle is an oblate ellipsoid when $D < 1$. The smaller $D$ is, the greater the degree of non-sphericity. The particle is an elongated ellipsoid when $D > 1$. There is a greater degree of non-sphericity

the larger $D$ is. The shape scale parameter of the rectangular shape (the bottom side is square) is the ratio of the height and the length of the bottom side. The degree of non-sphericity is the smallest when $D=1$, and the degree of non-sphericity



increases gradually when $D$ tends to 0 or infinity. The non-sphericity of ellipsoidal and rectangular with the same $D$ is somewhat different, depending on the shape of the model.

### 2.3 Aerosol mass concentration inversion method

The aerosol mass concentration is defined as the mass of dry aerosol particulate matter per cubic meter of air. According to the definition of aerosol mass concentration, the mass concentration $M(z)$ can be expressed as Eq. (4) (Li et al., 2021):

$$M(z) = \int_{r_{min}}^{r_{max}} \frac{4}{3}\pi r^3 \rho(r)n(r)dr \tag{4}$$

In which, $r$ is the radius of aerosol particles. To be consistent with the principle of DDA calculation, $r$ is the radius of the equivalent spherical shape of non-spherical particles. $\rho$ is the aerosol density, and the aerosol density of dust is 2.6μm/m³(Li

et al., 2021). Therefore, $\rho = 2.6$μm/m³ is set in this study. $n(r)$ is the APSD. By using the regularization method, the APSD is obtained by inverting two extinction coefficients and three backscattering coefficients (Müller et al., 1999). The functional is shown in Eq. (5).

$$g_i = \int_{r_{min}}^{r_{max}} K_i(m,r,\lambda_i)n(r)dr \tag{5}$$

$g_i$ is the extinction and backscattering coefficient detected by the lidar. $K_i$ is the kernel function. The functional relationship

between the $\alpha(z)$ and $n(r)$ is shown in Eq. (6) (Córdoba-Jabonero et al., 2019):

$$\alpha(z) = \int_{r_{min}}^{r_{max}} Q_{ext}(r,m,\lambda)\pi r^2 n(r)dr \tag{6}$$

$Q_{ext}$ is the extinction efficiency factor, which is calculated by DDA. $\lambda$ is wavelength, $m$ is complex refractive index, $r$ is particle radius. The extinction efficiency factor is calculated by DDA. Thus the aerosol mass concentration can be expressed as a function of the extinction coefficient as an independent variable (Ackermann et al., 1998). When incorporating the

extinction coefficient into the mass concentration formula, the dust aerosol particle mass concentration can be expressed as the particle mass extinction efficiency $K(z)$ and the extinction coefficient $\alpha(z)$. As shown in Eq. (7) and Eq. (8), the aerosol mass concentration can be obtained.

$$M(z) = \frac{\alpha(z)}{K(z)} \tag{7}$$

$$K(z) = \frac{\int_{r_{min}}^{r_{max}} \pi r^2 Q_{ext}(r,m,\lambda)n(r)dr}{\int_{r_{min}}^{r_{max}} \frac{4}{3}\pi r^3 \rho(r)n(r)dr} \tag{8}$$



Fig. 2 shows a flowchart of the present method. The near-surface data are obtained by Aerodynamic Particle Sizer (APS). The inversion of dust aerosol mass concentration in the high altitude is monitored by lidar and calculated with the DDA method.

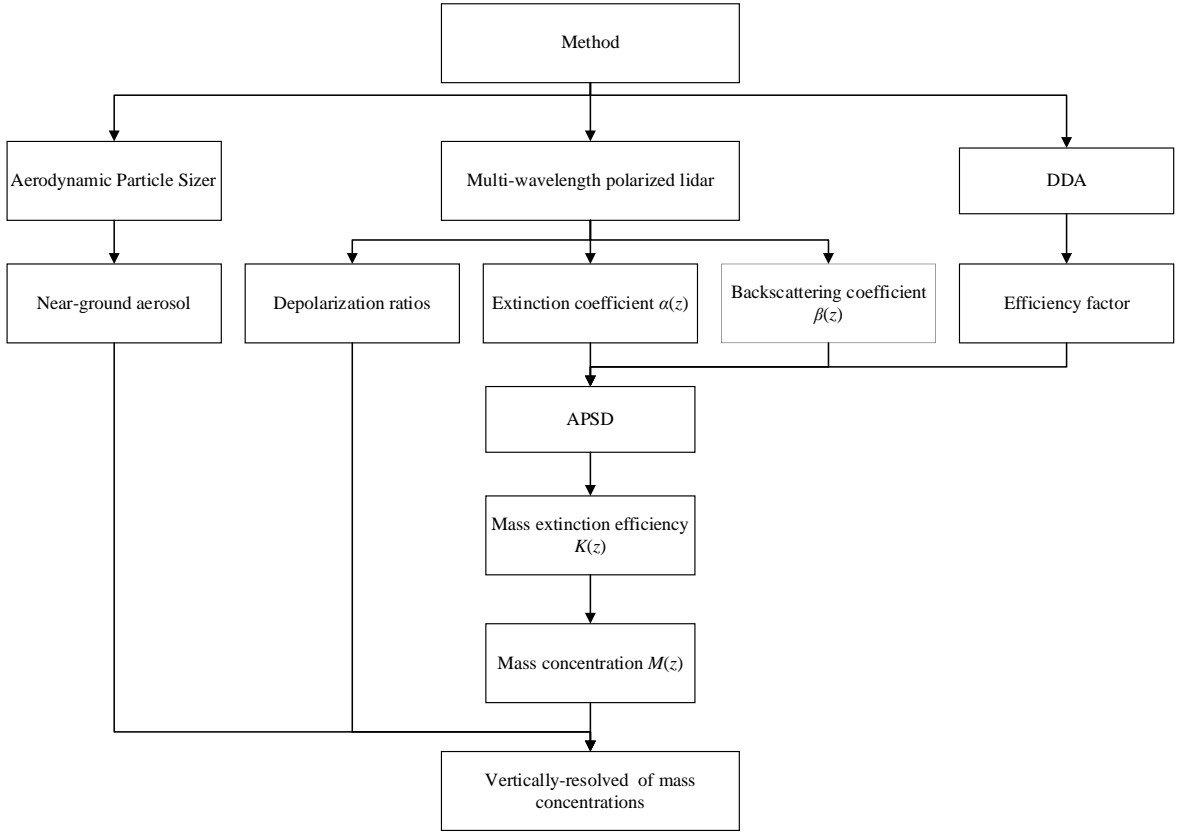

**Figure2: Flowchart of the mass concentrations inversion method**

**3 Experimental results**

A strong dust storm occurred in March 2021 in Yinchuan. Multi-wavelength polarized lidar was used to detect the vertical profile of optical parameters of dust aerosol. Since the extinction coefficient can reflect the degree of attenuation of light by suspended particles in the atmosphere. The depolarization ratio can well reflect the morphological characteristics of aerosol particles. Thus, the detected signals undergo inversion in order to derive the depolarization ratio $\delta_{532}$ and the extinction

coefficient $\alpha_\lambda$ at three different wavelengths (355nm, 532nm, and 1064nm). Fig. 3 displays the results of the calculation of the depolarization ratios and the extinction coefficients profiles of dust aerosol on March 16, 2021, at 2:01, 7:28, and 12:36.





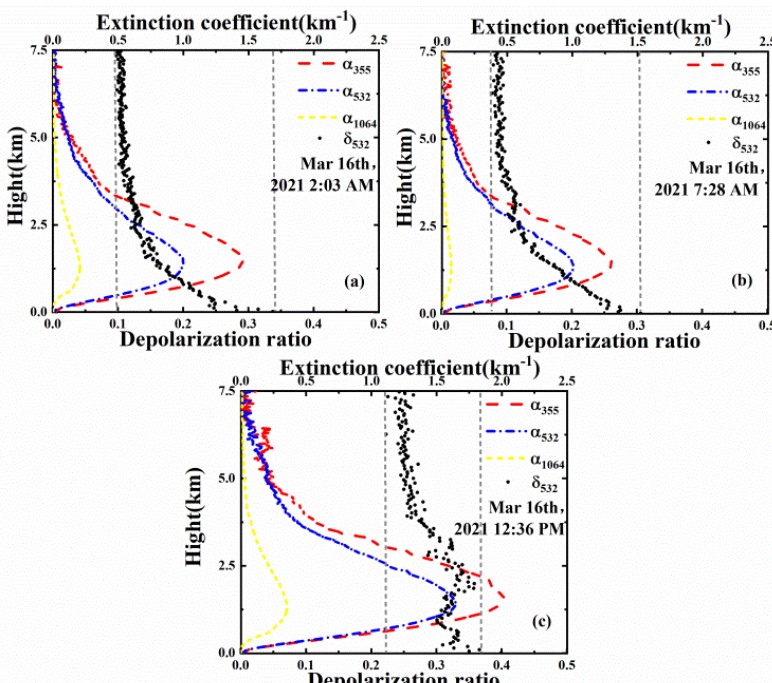

**Figure 3. Depolarization ratios and extinction coefficients of dust aerosols**

In Fig. 3, due to the overlapping area of the field of view, and the extinction coefficient and depolarization ratio at 0-1.6km

altitude are not for reference. Fig. 3(a), (b), and (c) represent the dust aerosol at 2:01, 7:28, and 12:36 on March 16, 2021, respectively. At wavelengths of 1064nm, 532nm, and 355nm, the extinction coefficients are represented by the yellow, blue, and red lines. The black dots represent the depolarization ratios detected by the 532nm polarization channel. Typically, larger particles scatter and absorb light more efficiently, with greater extinction coefficients. The extinction coefficient decreases with height due to the accumulation of aerosols in the near-ground region. In Fig. 3(a), 3(b), and 3(c), the

maximum values of extinction coefficients of 355nm are all larger than 1.0km$^{-1}$, which is consistent with the characteristics of dusty weather. The closer the depolarization ratio is to 0, the closer the aerosol particles are to a spherical shape. The closer the value of the depolarization ratio is to 1, the more serious the non-spherical characteristics of aerosol particles are. In Fig. 3(a), 3(b), and 3(c), the depolarization ratios of aerosols in this observation are all larger than 0.08. Especially at the time of 12:36, the depolarization range of dust aerosol particles is observed to be 0.22 to 0.37, which is typical of non-

spherical characteristics. Fig. 3(a), 3(b), and 3(c) show that the depolarization ratio is larger near the ground than the overall value at high altitudes. This results from massive dust aerosol particle accumulation close to the ground. It is worth noting that the extinction coefficient and the receding polarization ratio both show obvious bulges near 2-3km. This represents that the extinction values and particle shapes of the aerosol here have obvious characteristics of dust aerosols, and it also indicates that there is an aggregation layer of dust aerosols here.



From Fig. 3(a), 3(b), and 3(c), we can clearly recognize that the dust aerosol has non-spherical characteristics. It is not appropriate to use the Mie method to calculate the optical parameters such as the efficiency factor. To compute the optical properties of non-spherical particles, we use DDA in order to reduce the error. In order to better observe the difference in the extinction efficiency factor of non-spherical particles between different wavelengths and shapes. The wavelengths are set to 355nm and 1064nm. According to previous studies, dust aerosol particles were modeled or observed as approximately

ellipsoidal and rectangular (Huang et al., 2023; Horvath et al., 2016; Ulrich et al., 2005; Li et al., 2016). The DDA method is not only applicable to rotationally symmetric particles, but also to non-rotationally symmetric particles, so the DDA method is chosen in this study to calculate the light scattering characteristics of ellipsoidal and rectangular particles. The shapes are set to spherical, rotationally symmetric - ellipsoidal and non-rotationally symmetric - rectangular. In this study, the CRI of dust aerosols was set to 1.51+0.002i (Liu et al., 2008). Fig. 4 shows the extinction efficiency factors of non-spherical

particles with different shapes and wavelengths.

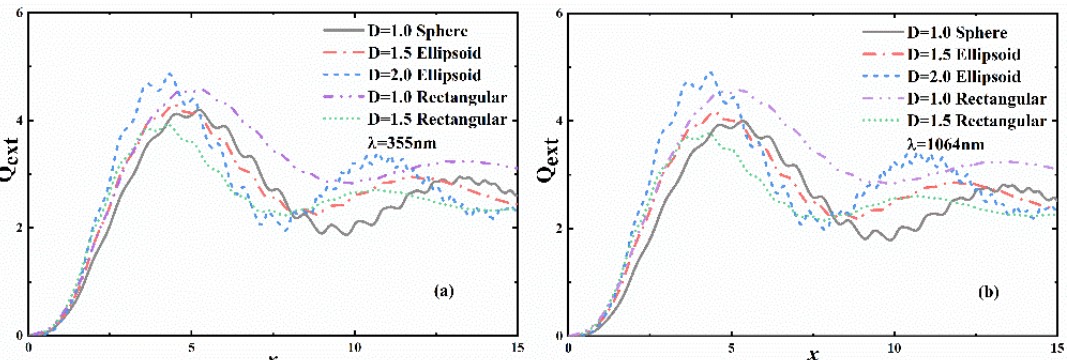

**Figure 4. Extinction efficiency factor for non-spherical particles of 355nm and 532nm wavelengths**

The $x=2\pi r/\lambda$ is aerosol size parameter. The spherical shape is the gray line. Elliptical shape with D=1.5 is represented by the red line. The blue line is the ellipsoidal shape with $D$=2.0. The purple line is rectangular for $D$=1.0. The green line is the

rectangular with $D$=1.5. It can find from Fig. 4 that when the size parameter $x$ increases, the extinction efficiency factors of all non-spherical particle forms exhibit an oscillating decreasing trend. The oscillation amplitude and period of the extinction efficiency factor of rectangular particles are smaller than those of ellipsoidal particles. The extinction efficiency of ellipsoidal particles shows an increase in oscillation amplitude and a decrease in period with the increase of the shape parameter $D$. The oscillation trends are similar between different wavelengths. However, the oscillation of the extinction

coefficient at 1064nm is slightly larger than at 355nm. The difference between the extinction coefficients of different shaped particles is also large. This indicates that the non-spherical aerosol particles' scattering are somewhat influenced by the wavelength and shape factors.

We calculated the dust aerosol mass concentration of ellipsoidal particles at different wavelengths $D = 1.5$ at noon on March 16, 2021 using this method. The wavelengths were set to 355nm, 532nm, and 1064nm. 1.51+0.002i was the complicated

refractive index. Fig. 5 displays the computation's results:



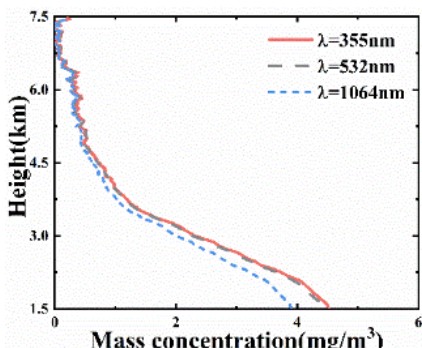

**Figure 5. Mass concentration of dust aerosols at different wavelengths**

In Fig. 5, the red, gray, and blue lines have respective wavelengths of 355nm, 532nm, and 1064nm. The values of all three mass concentrations decrease with increasing altitude. The main reason may be that the lower atmosphere is affected by human live as well and the dust aerosol undergoes deposition. Below 4km, the mass concentration at 355nm is more than that at 532nm and 1064nm. It is possible that on the way to a dust storm, a large number of fine particles are wrapped up in the wind. These fine particles arrive in the area and gradually settle down. The difference is especially obvious below 2km. However, the mass concentration curves of three wavelengths above 4km are close to each other, indicating that the number of particles of different sizes above 4km is relatively consistent.

Due to Eq. (9), it is known that the aerosol mass concentration is positively correlated with the extinction coefficient. We conducted a linear fit to the association between the extinction coefficient and mass concentration with the aim to confirm the accuracy of the mass concentration computation. 355nm, 532nm, and 1064nm were chosen as the wavelengths. Fig. 6 displays the results.

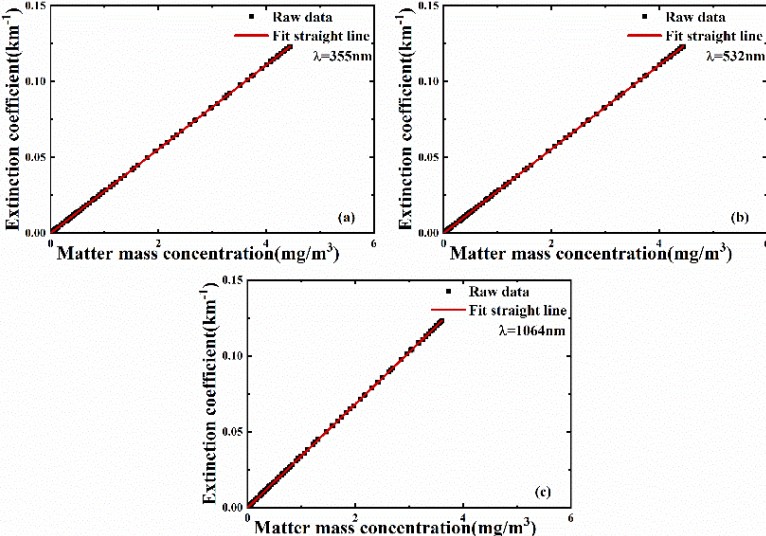

**Figure 6. Correlation of extinction coefficients and mass concentrations of dust aerosols**




A good linear relationship between the two can be found in Fig. 6. It shows that the trends of extinction coefficient and mass concentration are in accordance with a positive correlation. This method is applicable for calculating the mass concentration of dust aerosol particles.

To compare the impact of various particle shapes on mass concentration, the shape parameters were set as *D*=1.0 and *D*=1.5
rectangular, and *D*=1.5 and *D*=2 ellipsoidal. In order to observe the effect of wavelength more clearly, the mass concentrations at 355nm and 1064nm are discussed. Fig. 7 displays the computations' results.

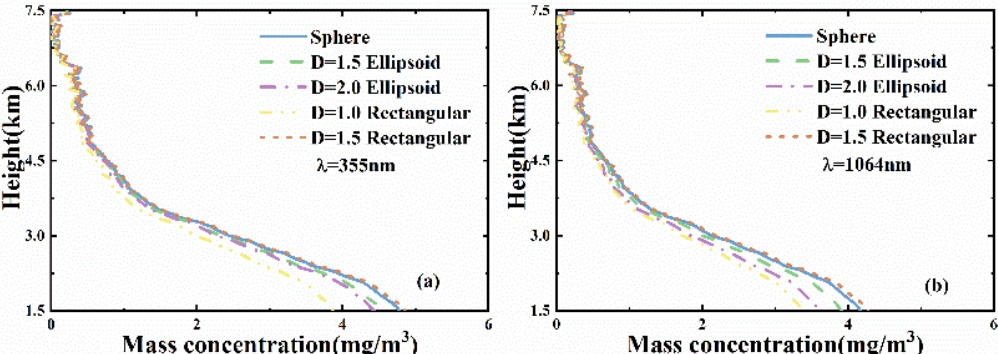

**Figure 7. Aerosol mass concentrations of dust aerosols at different shapes with 355nm and 1064nm**

The difference in mass concentration between spherical and non-spherical particles is discussed in Fig. 7. The mass
concentration of different shapes of particles decreases with the increase of height. The relative errors of rectangular with *D*=1.5 and spherical particles are 1.69% and 2.45% at 355nm and 1064nm, respectively. The difference in mass concentration between rectangular and spherical particles is the largest for *D*=1.0, with relative errors of 18.94% at 355nm and 19.23% at 1064nm, respectively. The effect of rectangular shape variation on the mass concentration is even greater than ellipsoidal. It can also be found that the larger the wavelength, the smaller the mass concentration.

In order to give a more complete picture of the overall change in mass concentration, we calculated the mean mass concentration at 355nm and 1064nm, and the results are displayed in Fig. 8(a). The wavelength is sensitive to the size of the particles, so the difference between wavelengths can reflect the difference in the number of particles of different sizes. The mass concentration difference between 355nm and 1064nm is shown in Fig. 8(b).

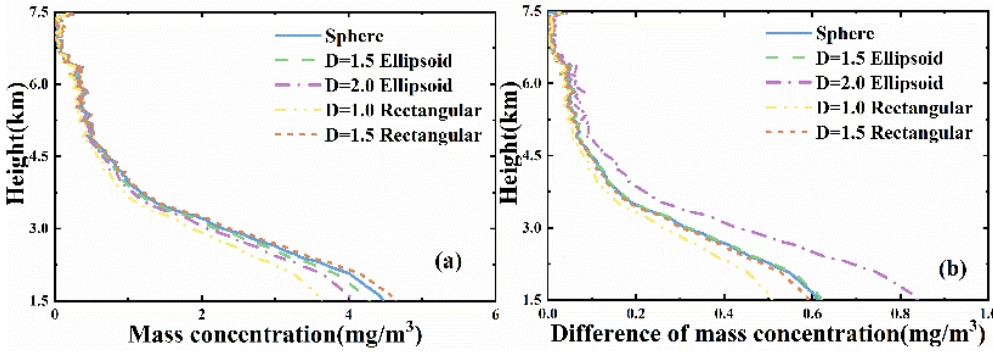



**Figure 8. Mean and difference of dust aerosol mass concentrations of different shapes at 355nm and 1064nm**

Fig. 8 compares the mean and difference in mass concentration at different wavelengths between spherical and non-spherical particles. Fig. 8(a) shows that the variation of mean mass concentration is consistent for different shaped particles. Similar to Fig. 7, the largest mass concentration difference is for the $D$=1.0 rectangular with a relative error of 19.08%. Fig. 8(b) shows that the largest difference between wavelengths is 0.84 mg/m$^3$ at the ellipsoid with $D$=2.0. This indicates that the largest

difference in the number of particles between large and small particles is found for the ellipsoid with $D$=2. This difference is obvious even at an altitude of 5-6.5km. Between wavelengths the smallest difference is the rectangular shape with $D$=1.0 which is 0.51mg/m$^3$. The mass concentration of spherical shape is close to the mass concentration of rectangular and ellipsoidal with $D$=1.5. The mass concentration of ellipsoidal particles is more easily affected by wavelength than rectangular particles.

It is not yet possible to directly determine the shape of dust aerosols by multi-wavelength polarized lidar detection. In view of the above findings, if we choose the rectangular shape to invert the dust aerosol may cause large unknown errors. We choose ellipsoidal particles to invert dust aerosol mass concentration. And because the maximum difference in mass concentration between different shapes is minimized by $D$=1.5 ellipsoidal shape. It can be seen from Fig. 8(b) that the difference in mass concentration of ellipsoidal particles with D=1.5 at different wavelengths is the smallest. Therefore, the

ellipsoidal shape with $D$=1.5 is used for the inversion of dust aerosol mass concentration in this study.

**Table 2 Complex refractive index of several aerosol types**

| Complex refractive index | Urban Industrial | Biomass Burning | Desert Dust and Oceanic |
|---|---|---|---|
| real part | 1.4-1.5 | 1.47-1.52 | 1.36-1.56 |
| imaginary part | 0.03-0.015 | 0.015-0.02 | 0.0015-0.003 |

The CRI is also an important parameter affecting aerosol mass concentration. Different types of aerosols have different ranges of real and imaginary parts of CRI as shown in Table 2 (Veselovskii et al., 2004). To discuss the impact of various

real and imaginary parts of CRI on the mass concentration and the maximum difference between different wavelengths, the maximum difference between different wavelengths was first calculated. According to Table 2, the real parts of CRI were set as 1.42, 1.45, 1.48, 1.51 and 1.54. To make it easier to observe the effect of the imaginary part on the mass concentration of dust aerosol, the imaginary parts of CRI were set as 0.002i, 0.005i, 0.008i and 0.01i. The results of the calculations of the difference in mass concentration between different wavelengths are shown in Fig. 9.



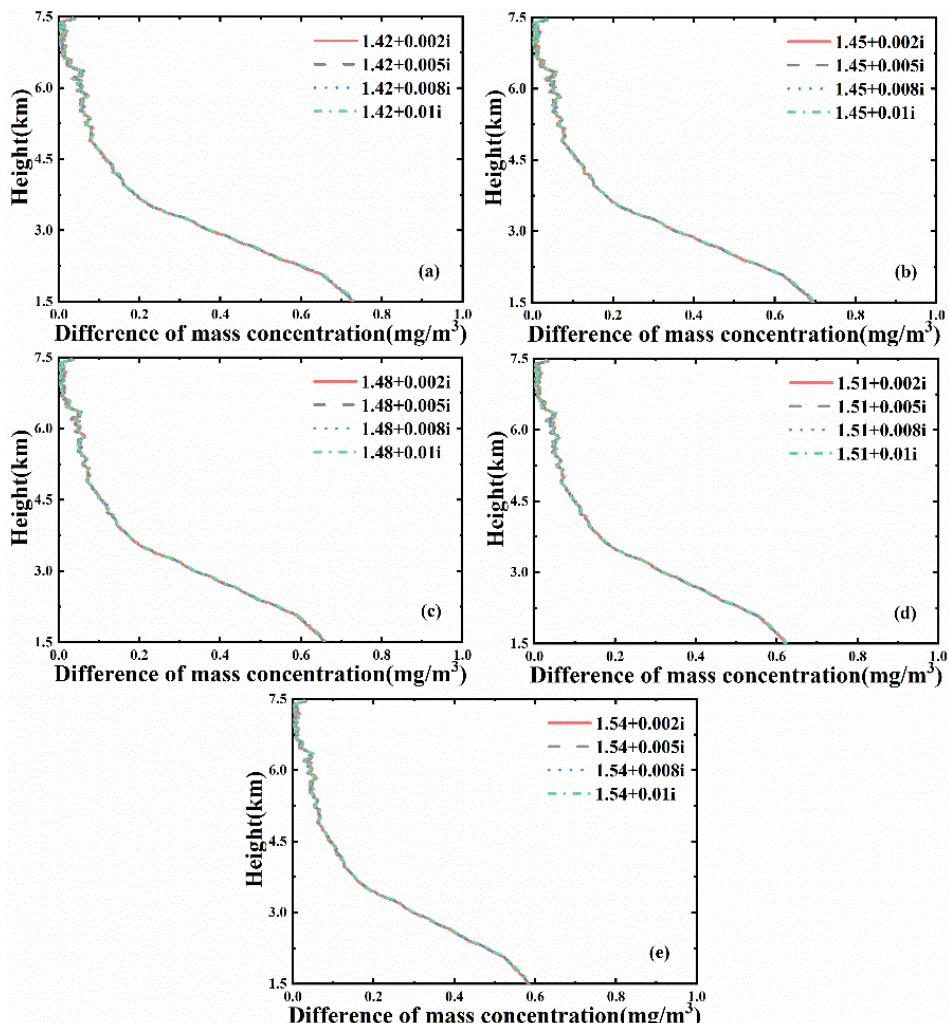

**Figure 9. Maximum difference in dust aerosol mass concentration with different imaginary parts**

Fig. 9 shows the variation of mass concentration difference for different imaginary parts for CRI real parts of 1.42, 1.45, 1.48, 1.51, and 1.54. It can be shown from Fig. 9 that as the difference of the imaginary part increases, so does the mass concentration difference. However, this difference is still small. And the larger the real part is, the smaller this difference is. Overall, the mass concentration difference between different imaginary parts almost coincides. This indicates that the impact of the CRI imaginary part on the mass concentration difference is small.

Since the difference in the mass concentration between the three wavelengths is small, in order to observe the effect of different CRI on the mass concentration more conveniently, we take the average value of the mass concentration of the three wavelengths to calculate, and Fig. 10 shows the results.





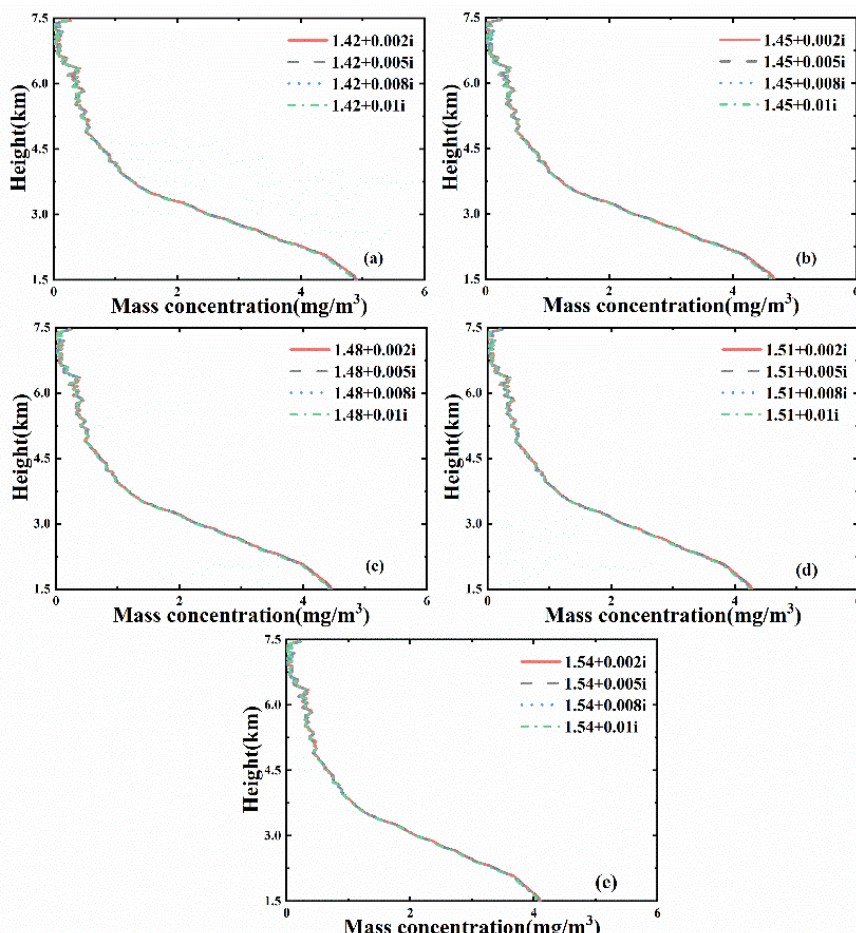

**Figure 10. Dust aerosol mass concentration mean value with different imaginary parts**

As can be seen in Fig. 10, the mass concentration profiles of different imaginary parts almost overlap. There is an insignificant difference in mass concentration only above 5km. However, this change becomes less and less obvious with the increase of the real part.

In order to discuss the effects caused by different real parts of CRI on the mass concentration and the difference between different wavelengths. The real parts of CRI were set as 1.42, 1.45, 1.48, 1.51 and 1.54. The imaginary parts of CRI were set as 0.002i, 0.005i, 0.008i and 0.01i. The results of the calculation of the difference in mass concentration between different wavelengths are shown in Fig. 11.




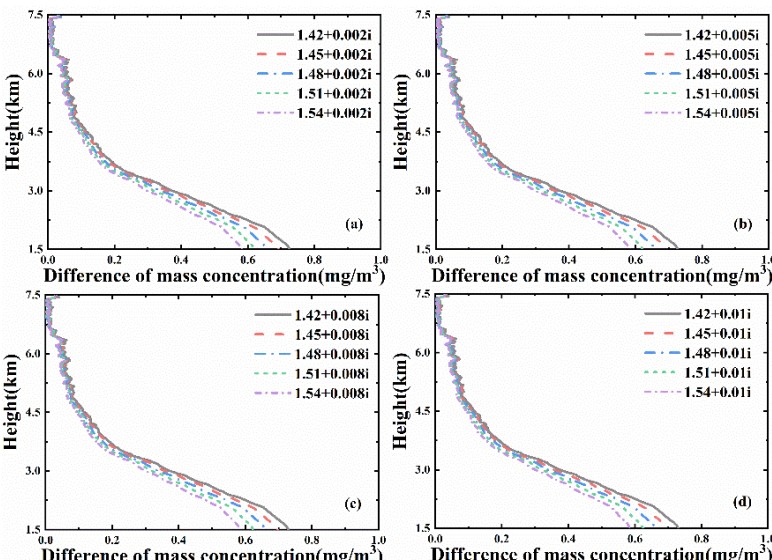

**Figure 11. Maximum difference in dust aerosol mass concentration with different real parts**

Fig. 11 shows the variation of mass concentration difference for different real parts for CRI imaginary parts of 0.002i, 0.005i, 0.008i, and 0.01i. It can be seen from Fig. 11, the mass concentration difference decreases with the increase of the real part with a negative correlation. When the imaginary part was 0.002i, the real part increased from 1.42 to 1.54, the maximum difference in mass concentration decreased by 0.1449 kg/m$^3$. When the imaginary part was 0.01i, the maximum difference in mass concentration decreased 0.1455 kg/m$^3$. Overall, the difference in mass concentration between different real parts is obvious. This indicates that the real part of CRI has a greater effect on the mass concentration than the imaginary part.

To observe the effect of different CRI real part changes on the mass concentration. We averaged the mass concentration for the three wavelengths. Fig. 12 shows the results of the calculation.

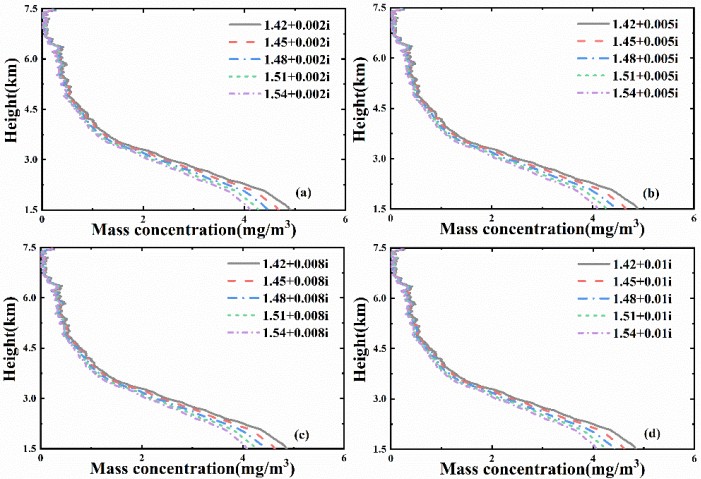

**Figure 12. Dust aerosol mass concentration mean value with different real parts**



From Fig. 12, it can be seen that there are some differences in the mass concentration of different real parts. The difference in mass concentration caused by different real parts is more obvious at 1.5km. The mass concentration decreases as the real part increases, with a negative correlation. When the imaginary part is 0.002i, the real part increased from 1.42 to 1.54, the mass concentration is reduced by 0.8001kg/m³. When the imaginary part is 0.01i, the mass concentration is reduced by

0.7811 kg/m³. Above 5km, there is only a small difference in the mass concentration profiles of different real parts. It is worth noting that the differences in the mass concentration profiles caused by the different real parts remain almost unchanged with the increase of the CRI imaginary part. Overall, the variation of the real part of the CRI has a greater effect on the mass concentration of particulate matter than the variation of the imaginary part.

To gain further insight into how the mass concentration of dust aerosol changes over time. We calculate the dust aerosol

mass concentration at different moments of this dusty weather. The CRI of spring dusty weather is 1.51+0.002i in Yinchuan (0). Therefore, in this study, the CRI was set to 1.51+0.002i. The non-spherical particle shape was set to an ellipsoid shape with $D$=1.5. In order to further study the aerosol number variations near the ground and at high altitude, the near-ground aerosol was monitored at this time using the APS. In Fig. 13, the computation and observation results are shown.

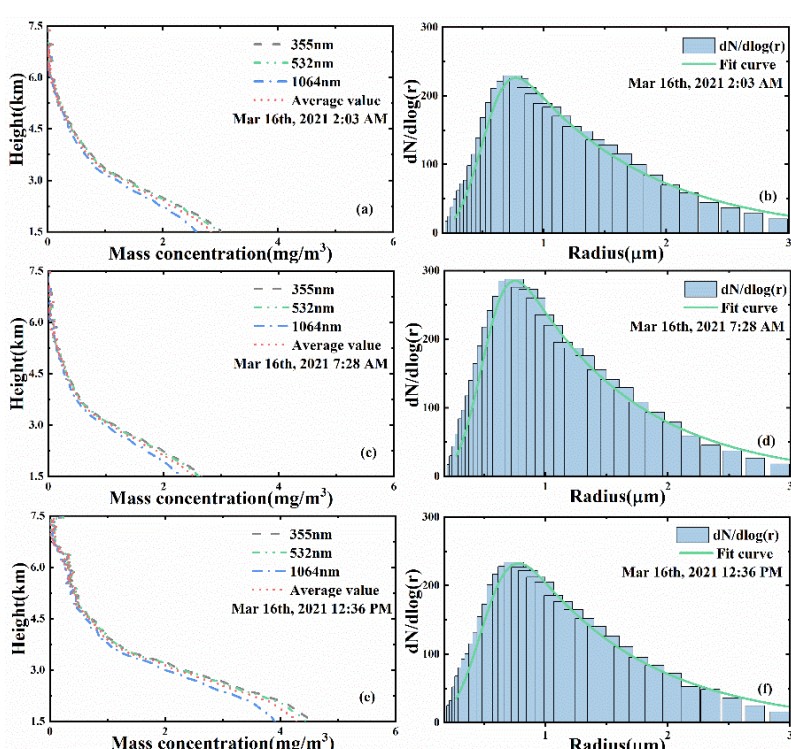

**Figure 13. High altitudes dust aerosol mass concentrations and near-ground dust aerosol number concentrations detected by the APS**

Fig. 13(a), (c), and (e) show the mass concentration inversion plots for March 16, 2021, at 2:03, 7:28, and 12:36 minutes, respectively. The gray line is 355nm, the green line is 532nm, the blue line is 1064nm, and the red line is the average of the



three wavelengths. The aerosol size distribution near the ground, as recorded by APS concurrently, is shown in Fig. 13(b),
(d), and (f). Fig. 13(a) shows that near the altitude of 1.5km, the aerosol mass concentration reaches a maximum value of
about 3.01mg/m$^3$ at 355 nm, 2.95mg/m$^3$ at 532nm, and 2.58mg/m$^3$ at 1064nm. The average value of the three wavelengths'
mass concentration is 2.83mg/m$^3$ at 2:03. Compared with Fig. 13(a), the mass concentration at all wavelengths in Fig. 13(c)
has decreased. Fig. 13(c) shows the average of the three wavelengths mass concentration at 7:28 is 2.51mg/m$^3$. Compared
with Fig. 13(c), Fig. 13(e) shows a significant increase in the mass concentration, which is higher than that of Fig. 13(a). Fig.
13(e) shows that at the altitude of 1.5km, the aerosol mass concentrations at 355nm, 532nm, and 1064nm reach the
maximum values of about 4.53mg/m$^3$, 4.44mg/m$^3$, and 3.91mg/m$^3$. The average value of the three-wavelength mass
concentration at 12:36 is 4.27mg/m$^3$. Fig. 13(a), (c), and (e) show a pattern of decreasing mass concentration with the
increase of altitude. Concentration gradually decreases with increasing height. However, in Fig. 13(e), a large amount of
aerosol still exists at a height of 4-7km. The reason for this surmise be the increase in the dust storm at this time. Observing
the mass concentration during this period with the mean value of the mass concentration at three wavelengths, it is found that
the mass concentration of dust aerosol at the high altitude undergo a process of decreasing and then increasing significantly.
At the same time, Fig. 13(b), (d), (f) show that the near-ground aerosol number changes show a clear pattern of growth
followed by a decrease.

On March 16, we reversed the mass concentrations in the atmosphere to further investigate the continuing variations in dust
aerosol mass concentrations near the ground and at high altitudes. Fig. 14 shows the results.

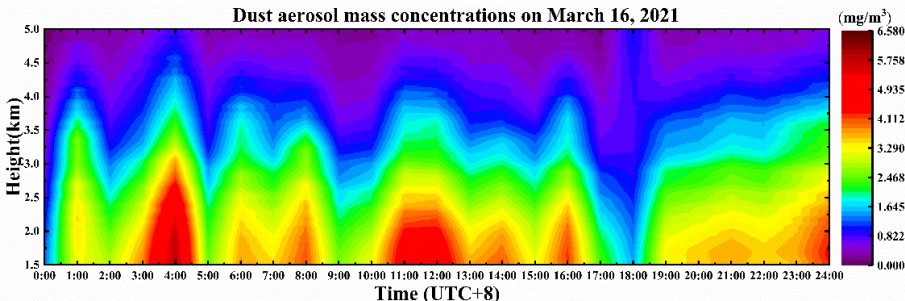

**Figure 14. Dust aerosol mass concentrations detected by the multi-wavelength polarized lidar**

Fig. 14 shows a gradual increase in mass concentration from 0:00 to 1:00 and a gradual decrease from 1:00 to 2:00. The
mass concentration increases rapidly from 2:00 to 4:00 and peaks at 4:00. At this time, the mass concentration value at
4.0km altitude is still large. This may be due to the fact that the airflow from the dust source was carrying a large number of
dust particles, which led to the aggravation of the local dust storm. However, the mass concentration decreased rapidly from
4:00 to 5:00. Winds bringing dust aerosols to other areas or sedimentation may have occurred during this time. Subsequently,
from 5:00 to 9:00, the mass concentration of atmospheric aerosol changed not much. In this period, the strong dust flow did
not have a great effect on the aerosol mass concentration. From 9:00 to 10:30, the mass concentration values peaked again
and continued consistently until 13:00. From 13:00 to 17:00, the mass concentration decreased fluctuating and reached a
minimum at 18:00. From 18:00 to 24:00, the mass concentration was once again in a continuous growth state. In general,



aerosol mass concentrations in the atmosphere change frequently when dust storms occur. This is due to the fact that changes in aerosol concentrations in the atmosphere are more susceptible to factors such as air currents than near the ground.

PM2.5 and PM10 of near-ground were monitored on March 16, 2021 at a monitoring station 5km from the lidar site as

shown in Fig. 15. The data is obtained from China Environmental Monitoring Center.

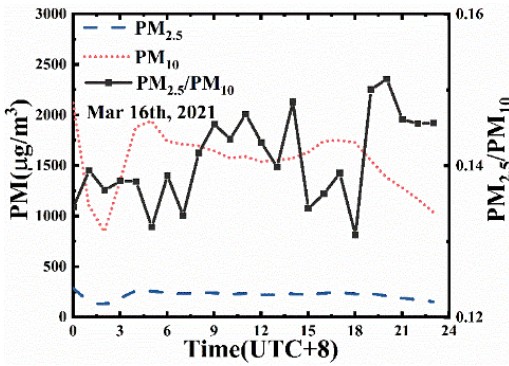

**Figure 15. PM2.5 and PM10 monitored at the site**

From Fig. 15, it can be seen that the number of large particles dominates at near-ground in this dust event. The trends of particle mass concentrations of PM2.5 and PM10 are similar, but the degree of change is more drastic for PM10. The PM10

drops sharply from 2118μg/m$^3$ at 0:00 to 841μg/m$^3$ at 2:00, which may be due to the strong airflow bringing back the near-ground aerosol particles to the high altitude, and then increases rapidly to 1941μg/m$^3$ from 2:00 to 5:00. Combined with Fig. 14, it also proves that such a strong dusty aerosol may be affected by the strong airflow from the dusty source. Thereafter, the mass concentration decreased slowly from 6:00 to 12:00. The dust state in the upper air is increasing volatility, and the dust aerosol near the ground is also decreasing slightly in this time period. Near-ground mass concentrations increased

slightly from 12:00 to 16:00. It is possible that the larger aerosol mass concentrations in the upper air were settling during this time period. Finally, at 18:00, the mass concentration of 1727μg/m$^3$ gradually decreases to 928μg/m$^3$. The mass concentration at high altitudes continues to increase during this time period, presumably because the winds are holding the aerosols near the ground and suspending them in the atmosphere. The concentration of PM10 at the surface is much higher than that of PM2.5. This shows that aerosol particles are mainly large particles during dust storm. As the PM10

concentration changes, the ratio of coarse to fine particles PM2.5/PM10 fluctuates between 0.151 and 0.131. The PM2.5/PM10 reached a minimum of 0.131 at 18:00 a.m. because PM2.5 was the minimum of the whole day. In general, the aerosol mass concentration changes more slowly near the ground than it does at high altitudes, as seen in Fig. 14. This may be due to the aggregation of large particles of aerosols near the ground, which are less affected by the airflow. There is a correspondence between the dust mass concentrations at high altitudes and near the ground at each time point of change.

This suggests that the dust mass concentrations at high heights and near the ground are correlated. From Fig. 14 and Fig. 15, it is hypothesized that the changes in aerosol mass concentrations near the ground and at high heights are consistent when affected by strong airflow from dust sources. When there is not affected by strong airflow from dust sources, the mass concentrations at high altitudes and near the ground may show an opposite trend due to local airflow and deposition.



## 4 Conclusions

In this paper, an inversion method based on DDA combined with multi-wavelength polarized lidar to calculate the vertical mass concentration of dust aerosol particles is proposed. First, based on the lidar signals, the extinction coefficient and backscattering coefficient of dust aerosols are inverted. Then the lidar's polarization channel is used to assess the dust aerosols' degree of non-sphericity. Considering that the aerosol depolarization ratio of observation reached as high as 0.37, the aerosol particles have obvious non-spherical characteristics. Therefore, DDA is used to determine the optical efficiency

factors of variously shaped non-spherical particles, different wavelengths, and different CRI. The results show that the extinction coefficient is significantly related to the mass concentration of the non-spherical particles with different shapes, wavelengths, and CRI, such as dust aerosol.

Based on the modeling of the optical properties of non-spherical aerosols, elliptical particles with $D = 1.5$ were used to invert the non-spherical aerosols of this dust event. The dust aerosol mass concentration is obtained by the mean mass

concentration of three-wavelength (355nm, 532nm and 1064nm). It is found by simulation calculation the impact of the imaginary part of CRI on the dust aerosol mass concentration is small. The real part of the CRI has a greater effect on the mass concentration than the imaginary part of the CRI. In order to better observe the change of the dust aerosol mass concentration as a dust storm develops. The mass concentration of dust aerosol at the time of the dust storm was inverted. We found that the three-wavelength mass concentrations at 2:03, 7:28, and 12:36 were 2.83mg/m$^3$, 2.51mg/m$^3$, and

4.2mg/m$^3$, at the altitude of 1.5km respectively. The APS and multi-wavelength polarized lidar were also used to observe the dust storms close to the ground and high-altitude simultaneously.

In this study, the relationship between changes in mass concentrations in the high altitude and near the ground is studied on the basis of lidar data and ground monitoring site during dust storms. It is found that dust aerosols are affected by gravitational sedimentation. The near ground dust aerosols were resuspended in the air due to wind and dust sources.

Changes in mass concentrations near ground and at great heights are correlated at the same moment. The dust aerosol mass concentrations at high altitude are more susceptible to factors such as air currents and change more frequently than aerosol mass concentrations near the ground.

The main advantage of the present method is that it overcomes the defect of the previous dust aerosol which adopts the spherical assumption. The multi-wavelength polarized lidar is used to detect the dust aerosols. It overcomes the temporal

discontinuity when relying on aerosol network observations. The present method can be applied to the inversion of dust aerosol mass concentrations in the atmospheric study. This helps us to further understand the evolution process of dust aerosols and dust storms. The study of dust aerosols mass concentrations can also provide support for pollution control in cities. It is of great significance for maintaining human health and protecting sustainable ecological environment.

**Acknowledgment**

The National Natural Science Foundation of China (NSFC) provided funding for this study (grants 62465001 and 42165010).

**Competing interests**

The authors declare that they have no known competing financial interests or personal relationships that could have



appeared to influence the work reported in this paper.

## Author contribution

Hu Zhao: Conceptualization, Methodology, Funding acquisition, Project administration, Writing – review & editing

Ze Qiao: Data curation, Investigation, Software, Writing – original draft

Jiyuan Cheng: Validation

Jiandong Mao: Supervision

Chunyan Zhou: Validation

Xin Gong: Validation

Zhiming Rao: Validation

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
