# Peer review of "Inversion of vertical mass concentration of non-spherical aerosols using multi-wavelength lidar"

_EGUsphere, 2025_

## Referee Comment (RC1)

The manuscript combines the regularization inversion and discrete dipole approximation (DDA) to retrieve height-resolved dust mass concentration from multiwavelength lidar measurements. The mass concentration is calculated from the aerosol particle size distribution (APSD) retrieved with the regularization method and the assumed constant particle density. The influences of several different non-spherical shapes are studied and the spheroid with an axis ratio of 1.5 was selected to represent the dust morphology. The overall structure is clear. However, I have major concerns whether this manuscript fits to a publication in AMT which I will describe in details in the following.

1. The study is neither substantially motivated nor sufficiently validated. The main highlight of this study is the consideration of dust non-sphericity in the size distribution retrievals. However, the authors didn't discuss what kind of errors will it bring if the conventional spherical assumption is used, which is viewed as the major motivation of this study. The accuracy of the method proposed in the manuscript is not properly discussed, either. The method is directly applied to real dust observations without any uncertainty analysis based on data simulation. There is also a lack of assessment of retrieval uncertainty associated to errors in the lidar-measured optical properties (particularly given the Klett method used in this study which introduce ambiguity due to the assumption of lidar ratio), the assumption of CRI and particle shape. Although there is discussion on the influence of exploiting different CRIs and particle shapes on the retrieval results in Sect. 3, without comparison with independent observations, it helps little in determining the optimal shape configuration and characterizing the associated uncertainty. The comparison with the ground in situ measurements is not discussed appropriately with consideration of related work and references.

2. The main conclusion that the proposed method overcomes the defect of previous methods adopting spherical assumption of this manuscript (Lines 378-379) is not convincing enough. It lacks evaluations of the accuracy of the retrievals obtained by applying both the previous method and the new method to the same observations. It is not convincing enough to select a largely simplified shape model (spheroids with a fixed axis ratio) as the shape configuration, compared to more complex but well-established models (e.g., spheroidal model by Dubovik et al. (2006) and irregular-hexahedral by Saito et al. (2021)).

3. Lots of statements are not true or at least not scientific rigorous (e.g., Lines 33-35, Lines 48-50). One of the most confusing points is the wavelength dependence of the mass concentration shown in the manuscript, which is not true by its nature definition. It could be related to lidar uncertainties in different wavelength channels, but this difference is never defined by the authors. In addition, the interpretation of Figure 4 is problematic and Figure 10 and Figure 12 are exactly the same (more details in the specific comments).

4. English expression is not fluent and precise enough. There are many repetitions (e.g., Lines 127-129), contradictions (e.g., Lines 108-113) and confusions (e.g., Lines 237-238).

With the concerns above, I suggest a rejection by AMT, or a major revision is possible if the authors are able to address these concerns.

Below are some specific comments that the author might consider.

**General comments**

1. The English expression of the manuscript can be improved. There are a few parts hard to read, logically confusing and even conflicting with each other. Please check them thoroughly.

2. Please make sure that all the parameters appearing in the equations are well defined.

3. In terms of parlance, is it "ellipsoid" or "spheroid"? Is it "cuboid/cube" or "rectangular"?

**Specific comments**

Lines 15-18: "It was found that in the detection of aerosol mass concentration by multiwavelength lidar, the larger the wavelength, the smaller the aerosol mass concentration. It was also found that…"
- How does the mass concentration depend on the measurement wavelength and the refractive index? Do you mean the retrieved mass concentration differs when different wavelengths and refractive indices are considered?

Lines 33-35: "The Mie method… is obviously not suitable".
- This statement does not hold all the time. For instance, the particle non-sphericity may have less impact on optical properties of small particles than big particles; and dust single scattering albedo may have little sensitivity to particle shape. Please justify it more rigorously with proper references.

Lines 48-50: "Point measurement methods… spatial variation of aerosols mass concentrations."
- Do you mean "in situ" measurements by point measurement? You state that the measurement "cannot acquire the spatial variation of aerosols mass concentrations" and it is not true exactly. Spatial variation can be acquired through airborne in situ measurements.

Line 63: "Additionally, the APSD was reversed using the regularization method."
- Do you mean "retrieve"?

Lines 66-68: "This paper also discusses… complex refraction index (CRI)."
- The shape and wavelength are not the "optical properties" of non-spherical particles, and the CRI is independent of particle shape, isn't it? Please improve this sentence.

Line 85: "The number of dipoles N=63461."
- Why this number? For what case is this setting suitable?

Eq. (1):
- Please define all the parameters in this equation.
- The letter "α" has already used for the expression of the extinction coefficient. Please consider another letter.

Lines 97-101 including Eq. (3):
- This part is very confusing and very hard to read. Please rephrase.

Lines 101-102: "Considering the radius of dust aerosol…"
- You need more proves to demonstrate this assertion. The paper of Müller et al. (1999) doesn't discuss anything about dust particle radius, why do you cite this paper? In addition, plenty of in situ measurements as well as sunphotometer

inversions indicate dust particles predominate in the range 1-10 μm. In some field measurements, freshly emitted dust even presents a super micron mode (e.g., Ryder et al., 2013).
- Thus, a more sufficient discussion is in need for supporting the setting of your simulation.

Lines 108-113:
- Fig. 1 and the following discussion refer to the spheroid rather than the ellipsoid. Please check the respective definitions.
- Looking through L108-110, does a smaller or a larger D represent a greater degree of non-sphericity exactly?
- "Fig. 1 illustrates that when D = 1, it is spherical." This is not consistent for a hexahedron (or "rectangular" in your expression). Apparently, when D = 1, it turns into a cube rather than a sphere.

Lines 118-119: "… r is the radius of the equivalent spherical shape of non-spherical particles."
- How is the "equivalent spherical shape" defined (e.g., the sphere being of the same cross section/surface area/volume/circumscribed radius… as the non-spherical particle)?

Eq. (6):
- For completeness, pleas also add the definition of backscattering coefficient.

Lines 128-137, Eq. (7-8) and Fig. 2:
- By "independent variable" (Line 129) and Eq. (7-8), do you mean the mass concentration can be calculated from the measured extinction? I don't think so since $K$ depends on APSD, CRI and particle shape factor. How do you decide $K$?
- Combining Eq. (7), (8), (6) derives Eq. (4), which signifies $M$ is neither dependent of $\alpha$ nor dependent of $\lambda$. What's the point to introduce Eq. (7) and (8)?
- For Fig. 2, (1) what is the function of "Depolarization ratios" in the method? (2) Can $K$ be uniquely determined by APSD considering it also depends on CRI?

Lines 154-156: "In Fig. 3(a), 3(b), and 3(c)… is consistent with the characteristics of dusty weather."
- High α355 is not sufficient evidence for dusty weather because (1) an increasing extinction coefficient can be caused be either an increase of particle size as the number concentration keep constant, or an increase of the number concentration; (2) due to the relatively larger size, dust particles influence the extinction less in the UV. In fact, looking at Fig. 3, I suspect 3(a), 3(b) represent the background aerosols having small depolarization ratio (0.1-0.15). Please verify my comment.

Fig. 4 and Lines 184-187:
- Both the figure and the following discussion are problematic. Instead of particle radius, you plot Qext against x, which is always the same at different wavelengths. So, there is no difference between 4(a) and 4(b). Thus, how do you get the result that "However, the oscillation of the extinction coefficient at 1064nm is slightly larger

than at 355nm." and the conclusion that "the non-spherical aerosol particles' scattering are somewhat influenced by the wavelength…"?

Lines 188-199 & Fig. 5:
- I am really puzzled about the dependence of the mass concentration on the wavelength presented in your results. How could it be since it means the mass of the particles per unit volume? Does it change with the wavelength? Which formular you use to calculate mass concentration?
- In Lines 195-197, do you mean that the "higher" value at 355 nm compared to other wavelengths is caused by larger fraction of fine particles? What is the exact driving mechanism? In addition, I doubt that the potential "fine particles" are more local than transported with dust inasmuch as they are harder to deposit compared to the coarser dust particles?
- In Lines 198-199, could it result from the loss of retrieval sensitivity and SNR as the particles are diluted with the increase of altitude?
- What is the uncertainty of the mass concentration retrieval?

Lines 200-208 & Fig. 6:
- The logic of this part is problematic. You cannot draw the conclusion that the mass concentration is positively correlated with the extinction coefficient from Eq. (8) because apparently, $K$ is a function of $\lambda$, CRI and APSD. It is possible that the two quantities show a perfect linear relation if only the number concentration varies with altitude, while both CRI and APSD shape keep constant, which is the case in this observation. However, this is not sufficient to demonstrate the accuracy of the mass concentration retrieval, which, according to Eq. (4), is essentially determined by the accuracy of the retrieved APSD and particle density.

Lines 235-236: "In view of the above findings, if we choose the rectangular shape to invert the dust aerosol may cause large unknown errors."
- The above results only show there are differences in the retrieval between the rectangular and ellipsoidal shapes, but why do you state that the rectangular shape may cause large unknown errors? Do you know the true value of mass concentration? Why couldn't be the ellipsoidal shape assumption that has large error? How to prove this?

Lines 237-238: "And because the maximum difference in mass concentration between different shapes is minimized by D=1.5 ellipsoidal shape."
- I do not really follow this sentence. The difference between different shapes should be derived by comparing different shapes. How do you define the "difference" by only looking at one shape (i.e., $D$ = 1.5 ellipsoidal shape)?

Line 280 (Fig. 12):
- What is the difference between Fig. 12 and Fig. 10?

Line 290: "The CRI of spring dusty weather is 1.51+0.002i in Yinchuan."
- Please add proper references or justify the rationality of this CRI value.

Lines 347-353:
- Please supplement this part with more synoptic proves and analyses to explain, for example, the formation of the "aggregation of large particles near the ground", the source of the dust observed by lidar (local? transported?), and the motion of the air flow when the lidar and ground observations behave consistently and oppositely. Otherwise, these "hypothesizes" may not sufficient to validate your lidar retrievals, particularly given the fact that the lidar retrievals and the ground APS measurements are not always consistent. Moreover, add more comparisons with other studies to support your conclusions.

Lines 373-374: "It is found that dust aerosols are affected by gravitational sedimentation. The near ground dust aerosols were resuspended in the air due to wind and dust sources."
- Solely comparison between lidar retrievals and ground measurements without further synoptic analysis is not enough to support these conclusions.

Lines 378-379: "The main advantage of the present method is that it overcomes the defect of the previous dust aerosol which adopts the spherical assumption."
- Throughout this paper, you do not mention in details what defects of the spherical assumption in terms of dust mass concentration retrievals. Do they lead to an over- or underestimation of the retrieval result? These are of significant importance by motivating the development of your new method.
- I cannot know from your study how "the present method overcomes the defect of the previous methods" and to what extent the present method improves the retrieval accuracy.
- I cannot know the performance of the present method because the retrieval accuracy is not sufficiently assessed. The discussion of the real case retrievals shows the potential influence of the CRI and particle shape on the retrieval result, but there is not a specific uncertainty report.
- You choose $D$ = 1.5 ellipsoid as the shape configuration. However, it is hard to verify it is the optimized configuration since you do not exploit a quantitative measure to evaluate the possible retrieval error when different configurations are set.

**Technical corrections**

Line 8: A method of inverting…

Line 75: The 532 nm polarized signals include… different particle radii.

Line 76: Therefore, the optical parameters at different wavelengths…

Lines 78 and 79 (Table 1): … of the multi-wavelength polarized lidar.

Line 89: where $\mathbf{P}_j$ … ("w" in lowercase)

Line 118: in which, …

Line 127 & Line 128: Why do you repeat this sentence twice?

Line 141-145: "Since the extinction coefficient … coefficient $\alpha\lambda$ at three different wavelengths (355nm, 532nm, and 1064nm)."

- It is better to rephrase them as one sentence.

Line 188: … at different wavelengths for D = 1.5…

---

## Community Comment (CC2)

Dear Professor Tetiana Kalinichenko,

Thank you very much. The suggestions you gave me for my article are very good and I have benefited a lot from them. I'm very glad to answer the questions you raised. The answers to each question are as follows:

1.  What unit of "aerozol density of dust" was used in your calculations? Please check 119-120 rows. I just inspected reference (Li et al., 2021) and didn't find it.

**Answer**: I'm very sorry. Due to my negligence, I wrote the unit as $\mu m/m^3$ and the references provided were also inappropriate, which caused you a misunderstanding. The unit of the **density** of dust aerosol in my article is $g/cm^3$, which has been verified in the following two references. I will cite these two references. In both of these two literatures, the unit of dust aerosol **density** is used as $g/cm^3$

**Reference1:** Veselovskii, I., Barchunov, B., et al.: Retrieval and analysis of the composition of an aerosol mixture through Mie–Raman–fluorescence lidar observations, Atmospheric Measurement Techniques, 17: 4137-4152, doi: 10.5194/amt-17-4137-2024, 2024.

Download:https://amt.copernicus.org/articles/17/4137/2024/amt-17-4137-2024.pdf

**Reference2:** Ansmann, A., Seifert, P., Tesche, M., Wandinger, U.: Profiling of fine and coarse particle mass: case studies of Saharan dust and Eyjafjallajokull/Grimsvotn volcanic plumes, Atmos. Chem. Phys., 12: 9399-9415, https://doi.org/10.5194/acp-12-9399-2012, 2012.

Download:

https://www.researchgate.net/publication/258564791_Profiling_of_fine_and_coarse_particle_mass_case_studies_of_Saharan_dust_and_EyjafjallajokullGrimsvotn_volcanic_plumes

2.  Am I correct in understanding that the dust mass concentration is expressed in $mg/m^3$? Many environmental and health-related studies commonly use $\mu g/m^3$, so I would appreciate confirmation to ensure accurate interpretation and comparison with other data sources.

**Answer**: Thank you for your very good suggestion. Your understanding is correct. In this article, the mass concentration of dust is expressed in $mg/m^3$ This paper studied the mass concentration of dust aerosol during a strong dust storm in March 2021 in Yinchuan. Therefore, the mass concentration was relatively high this time, and thus the unit of mass concentration, $mg/m^3$, was used. You're right. The commonly used unit for aerosol mass concentration is $\mu g/m^3$. In this paper, $mg/m^3$ is used. This shows that during a sandstorm, the mass concentration of dust aerosols in the atmosphere is many times higher than usual.

3.  I didn't find Equiation 9.

**Answer**: Thank you for your very good suggestion. I carefully checked the paper. It was my mistake. It should be Formula 8 (200 rows).

Sincerely,

Hu Zhao